# Assessing the Species in the CARES Preservation Program and the Role of Aquarium Hobbyists in Freshwater Fish Conservation

**Jose W. Valdez** [1] and **Kapil Mandrekar** [2,*]

1   Department of Bioscience—Biodiversity and Conservation, Aarhus University, Grenåvej 14, 8410 Rønde, Denmark; jose.valdez@bios.au.dk
2   Department of Environmental and Forest Biology, SUNY College of Environmental Science and Forestry, 1 Forestry Drive, Syracuse, NY 13210, USA
*   Correspondence: kapil.mandrekar@gmail.com

**Abstract:** Freshwater fish represent half of all fish species and are the most threatened vertebrate group. Given their considerable passion and knowledge, aquarium hobbyists can play a vital role in their conservation. CARES is made up of many organizations, whose purpose is to encourage aquarium hobbyists to devote tank space to the most endangered and overlooked freshwater fish to ensure their survival. We found the CARES priority list contains nearly six hundred species from twenty families and two dozen extinct-in-the-wild species. The major families were typically those with the largest hobbyist affiliations such as killifish, livebearers, and cichlids, the latter containing half of CARES species. CARES included every IUCN threatened species of Pseudomugilidae and Valenciidae, but only one percent of threatened Characidae, Cobitidae, and Gobiidae species. No Loricariidae in CARES were in the IUCN red list as they have not been scientifically described. Tanzania and Mexico contained the largest amount of species, with the latter containing the most endemics. Many species were classified differently than the IUCN, including a third of extinct-in-the-wild species classified as least concern by the IUCN. This vast disconnect exemplifies the importance of future collaboration and information exchange required between hobbyists, the scientific community, and conservation organizations.

**Keywords:** aquarists; aquarium trade; captive-breeding; IUCN red list; ornamental fish; threatened fish; undescribed species

## 1. Introduction

Although freshwater habitats constitute only 0.01% of all water on Earth and 2.3% of the Earth's surface, they support approximately 9.5% of all described animal species, including one third of all vertebrates [1,2]. Given the disproportionately high biodiversity value of freshwater systems, it is of serious concern given that they are one of the most threatened habitats on Earth. The World Wide Fund for Nature's Living Planet Index 2018 indicates that populations of freshwater species have declined by an average of 83% since 1970, much larger than declines seen in terrestrial (38%) and marine (36%) species [3]. This is especially true for freshwater fishes which the IUCN has named as the most threatened vertebrate group by number of species [4].

Freshwater fish make up approximately half of all known fish species and nearly a quarter of global vertebrate diversity, with many new species being discovered every year [4]. Despite marine ecosystems being comprised of a relatively larger area, fish species are far richer per volume in freshwater habitats. This is due to the geographical isolation of these systems, which has led to the evolution of many species with very small ranges that may encompass a single isolated lake or river basin [1,4]. Such high

levels of fragmentation and resultant high species endemism has made freshwater habitats biodiversity hotspots. However, it also makes them especially sensitive to anthropogenic impacts where large numbers of species can rapidly become extirpated [1]. Even for those species that have yet to entirely disappear, human activities have reduced or eliminated such a high proportion of populations that they have incurred an extinction debt due to their low-viability [1]. Currently, the greatest threats that freshwater habitats are facing include habitat degradation from pollution (contaminants, micro-plastics, and algal blooms), flow modification (dams and hydropower), overexploitation (commercial fishing), climate change, invasive species, and infectious diseases [3,5]. These combined, and often interacting, changes cause bottom-up and top-down ecosystem level changes, the net effect of which is a reduction in the future viability of freshwater species [3]. Expanding population pressures and accelerating urbanization, along with the ever-growing need for fresh water and food production, irrigation and water infrastructure developments, will only exacerbate the steep decline and loss of freshwater biodiversity [3,5]. It is the magnitude of these anthropogenic threats to freshwater fish that now warrants a more proactive and interventionist conservation strategy that combines different levels of management with a multi-stakeholder approach [4].

Various organizations can play important roles in the conservation of biodiversity and habitats. For example, zoological organizations such as the American Zoo and Aquarium Association (AZA), the European Association of Zoos and Aquaria (EAZA), and World of Zoos and Aquariums (WAZA) integrate a holistic approach to conservation which includes public education, field conservation, species survival plans, and ex-situ conservation programs [6,7]. Zoos invest more than US $350 million annually on captive breeding and reintroductions, which are the most frequently cited action leading to the most improvements on the IUCN Red List status of a species [8]. However, although the number of aquariums has increased at a much faster rate than that of zoos, with up to 450 million people visiting each year [6], they are vastly under-represented in conservation projects given their prevalence in nature and popularity [9]. Aquariums and zoos only hold about 7% of all threatened fishes with only two out of 31 actively involved in a fish conservation reintroduction project [9,10]. This may be attributed to the undefined conservation needs of many fish species, with approximately only half of all known species having been assessed by the IUCN [9]. To close these conservation and knowledge gaps, aquarium hobbyists, which make up 99% of the global ornamental fish market, may play a vital role given the considerable expertise of aquarists in the husbandry, reproduction, and ecology of fresh water species [11].

The global trade in ornamental fish has grown 14% annually since the 1970s and now involves approximately 125 countries [12,13]. Over 1 billion fish are internationally traded annually, including over 5300 freshwater and 1800 marine species, with an estimated worth of between US $15–30 billion each year [6,13,14]. This international trade is dominated by freshwater fishes, accounting for over 90% of the total trade volume, with 90% of species being captive bred and typically sourced from breeding facilities in Asia, South America, Israel, USA, and Europe [13–15]. Recently, fisheries and ornamental fish organizations have recognized their role in freshwater fish conservation by creating initiatives which promote sustainable practices that serve to provide a livelihood for local communities, promote environmental stewardship, and protect vulnerable freshwater ecosystems and species. Notable examples include Project Piaba, a community-based interdisciplinary project supported by zoos and aquariums which promotes sustainable fisheries and provides a livelihood for local communities [16]; the Indonesia Nature Foundation, which is among the main suppliers of the Indonesian fish trade and supports communities to build artificial reefs, trains in sustainable collection methods, and helped establish a captive rearing program for the Banggai cardinalfish (*Pterapogon kauderni Koumans*); and the AZA Freshwater Fish Taxon Advisory Group which helps conserve Lake Victoria-Kyoga's indigenous fishes including Lake Victoria cichlids, many of which were donated or acquired through hobbyist and the aquarium trade [4].

Although aquarium keeping is a popular hobby with millions of enthusiasts worldwide [17], serious aquarium hobbyists have considerably more passion and knowledge for fish compared to the

casual aquarium owner. These aquarists often own multiple tanks, are involved in captive-breeding, tend to be more interested in the science of fish and aquatic environments, and are typically affiliated with an aquarium hobbyist society/organization. Many of these aquarium hobbyist organizations have been instrumental in leading their own projects and generating scientific knowledge in collaboration with professional scientists. This includes the discovery of new species such as *Pseudolaguvia lapillicola*, *Danionella dracula*, the rainbow killifish (Nothobranchius rachovii), the La Luz Splitfin goodeid (*Zoogoneticus purepechus*), and the Sahara aphanius (*Aphanius saourensis*), as well as the rediscovery of the Azraq killifish (*Aphanius sirhani*) [12]. Many species have even only been described and named by the respective organizations based on physical characteristics and locations, since they remain unknown by researchers or neglected by governments and conservation organizations. Several hobbyist associations also maintain extensive database depositories of thousands of species (e.g., catfish (http://www.planetcatfish.com/), Fresh Water Fishes of Mexico (http://www.mexfish.info/default. php?lang=es), killifishes (http://www.killi-data.org), and cichlids (http://www.cichlidae.com/), with information on their biology, ecology, range, and behaviors. These hobbyist organizations take an active role in conservation projects funded by their members or crowdfunding, which is especially important for non-commercially important species, and have led to the successful reintroduction of the endangered Spanish toothcarp (*Aphanius iberus*) in restored lagoons by the Llobregat delta Sociedad de Estudios Ictiológicos (SEI), and three endangered Aphanius species (*A. apodus, A. danfordii and A. sirhani*), as well as the wild-extinct Potosi pupfish (*Cyprinodon alvarezi*) by the Spanish Killifish Association (SEK) [12]. Since many species (mainly livebearer, cichlids, and killifish species) are available only from aquarium hobbyists (approximately a quarter of aquarium species are exclusively owned by hobbyists), hobbyist conservation projects have been created to help maintain a viable bank of germplasm of the most endangered species [13]. This includes conservation projects such as the Fish Ark Project (FAP), Hobbyist Aqualab Conservation Project (HACP), and the Goodeid Working Group (GWG), which successfully keep populations of the 12 most endangered or extinct-in-the-wild and 24 threatened goodeid species in Mexico and have provided specimens of rare fishes to 34 universities, public aquaria, zoos, and other hobbyists in 15 countries to ensure species survival [12].

The largest conservation hobbyist organization is the CARES (Conservation, Awareness, Recognition, Encouragement, and Support) preservation program. Founded in 2004, CARES is currently made up of 30 aquarium societies and international organizations whose purpose is to encourage serious aquarium hobbyists around the world to devote tank space and distribute to other members one or more of the listed vulnerable, endangered, or extinct-in-the-wild species to help preserve them for future generations. Their other main goal is to share ecological, husbandry, and habitat knowledge about these species with other aquarists, scientists, and conservationists. The CARES program also has their own risk classification, conducted by professional scientists, of listed species not classified by the IUCN red list or those they believe require a different classification. However, to better serve its goals of preserving at-risk species and sharing information, it is critical to assess the species within the program to evaluate what taxonomic groups and regions are well represented and which may require more attention, as well as comparing their conservation classification to those of the IUCN. The aim of this paper is to assess the species and regions in the CARES preservation program, compare their IUCN conservation status, and provide suggestions for future directions.

## 2. Results

The CARES priority list contained 572 freshwater species in 20 different family groups with 30 species classified as extinct-in-the-wild. The priority list was overrepresented by Cichlidae with 47% of all CARES species (268), which also made up over half of the extinct-in-the-wild species (17) (Table 1). The families representing the highest proportion of the total IUCN threatened and data deficient species were Pseudomugilidae and Valenciidae, where all species were represented, and Goodeidae and Aplocheilidae with over 80% of the total threatened and data deficient species in the IUCN (Table 1). Approximately two-thirds (383) of the species were found in the IUCN red list,

82% (471) were in fishbase.org database, while 14% (85) were not found in either database (Table 1). The group with the highest proportion of unclassified species not identified in any database was Loricariidae with 58.6% (17), Cichlidae with 22.8% (61) and Aplocheilidae with 15.4% (2) (Table 1).

**Table 1.** Families in the CARES priority list and their IUCN classification.

| Family | CARES Total | Classified in IUCN | IUCN Total Threatened | Proportion of IUCN Threatened/Data Deficient | Undescribed | Critically Endangered | Extinct in the Wild |
|---|---|---|---|---|---|---|---|
| Adrianichthyidae | 6 | 6 | 15 | 0.40 | 0 | 0 | 0 |
| Anabantidae | 42 | 14 | 9 | 1.56 | 0 | 5 | 0 |
| Aplocheilidae | 13 | 8 | 10 | 0.80 | 2 | 2 | 1 |
| Bedotiidae | 23 | 21 | 28 | 0.75 | 2 | 5 | 0 |
| Characidae | 3 | 1 | 91 | 0.01 | 0 | 0 | 0 |
| Cichlidae | 268 | 197 | 500 | 0.39 | 61 | 60 | 17 |
| Cobitidae | 1 | 1 | 79 | 0.01 | 0 | 0 | 0 |
| Cyprinidae | 20 | 18 | 1081 | 0.02 | 1 | 2 | 1 |
| Cyprinodontidae | 16 | 16 | 44 | 0.36 | 0 | 5 | 4 |
| Fundulidae | 1 | 1 | 9 | 0.11 | 0 | 1 | 0 |
| Gobiidae | 3 | 3 | 254 | 0.01 | 0 | 0 | 0 |
| Goodeidae | 37 | 13 | 15 | 0.87 | 0 | 10 | 3 |
| Loricariidae | 29 | 0 | 62 | 0.00 | 17 | 0 | 0 |
| Melanotaeniidae | 22 | 18 | 26 | 0.69 | 0 | 0 | 2 |
| Mochokidae | 6 | 6 | 84 | 0.07 | 0 | 0 | 0 |
| Nothobranchiidae | 25 | 24 | 120 | 0.20 | 1 | 0 | 0 |
| Poeciilidae | 28 | 19 | 60 | 0.32 | 0 | 3 | 2 |
| Pseudomugilidae | 13 | 6 | 6 | 1.00 | 1 | 3 | 0 |
| Rivulidae | 14 | 9 | 36 | 0.25 | 0 | 0 | 0 |
| Valenciidae | 2 | 2 | 2 | 1.00 | 0 | 2 | 0 |
| Total | 572 | 383 | 2531 | | 85 | 98 | 30 |

Of those species in the IUCN, 64 (16.7%) species were classified differently than the IUCN, with seven species labeled least concern and that CARES classified as extinct-in-the-wild (Figure 1). Due to some species classified differently by CARES, Anabantidae had more threatened species than were classified as threatened in the IUCN (Table 1). None of the 29 Loricariidae species were found in the IUCN red list, while only 1% of threatened Characidae, Cobitidae, and Gobiidae species were in CARES (Table 1). The greatest number of species came from East Africa, Mexico, Brazil, Southeast Asia (Figure 2a), specifically Tanzania (118), Mexico (78), Madagascar (66), Kenya (61) and Uganda (60) (Figure 2a). The greatest number of endemics were from Mexico (76, 17.5%) and Madagascar (65, 15%), followed by Tanzania (47, 11%), Brazil (43, 9.9%), and Indonesia (42, 9.7%) (Figure 2b).

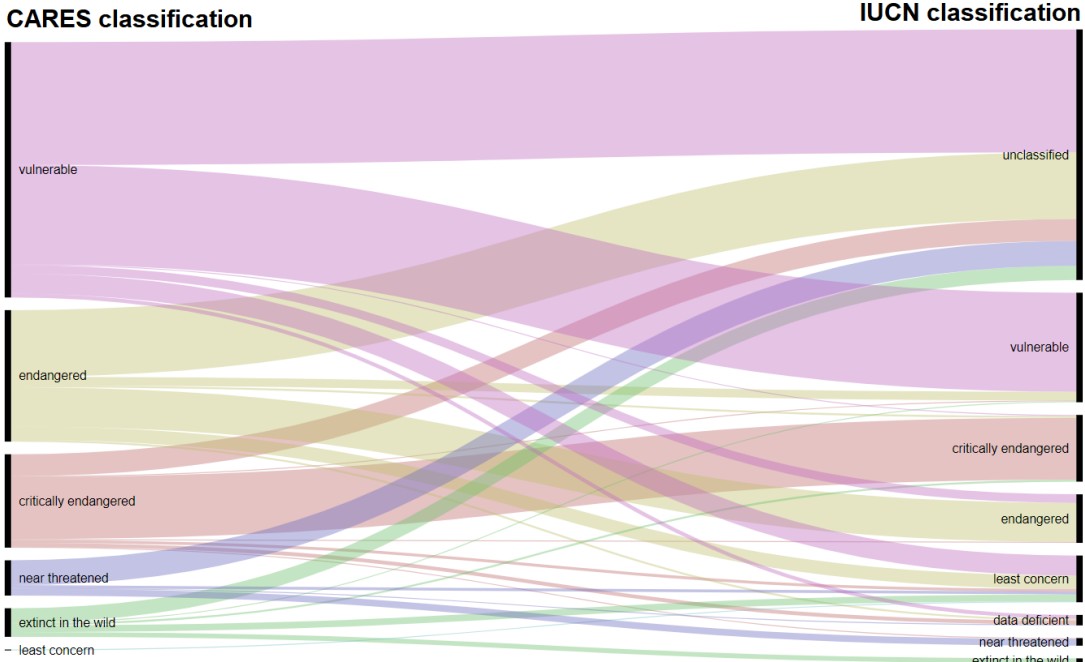

**Figure 1.** Classification of CARES species found in the IUCN. http://app.rawgraphs.io/http://app.rawgraphs.io/.

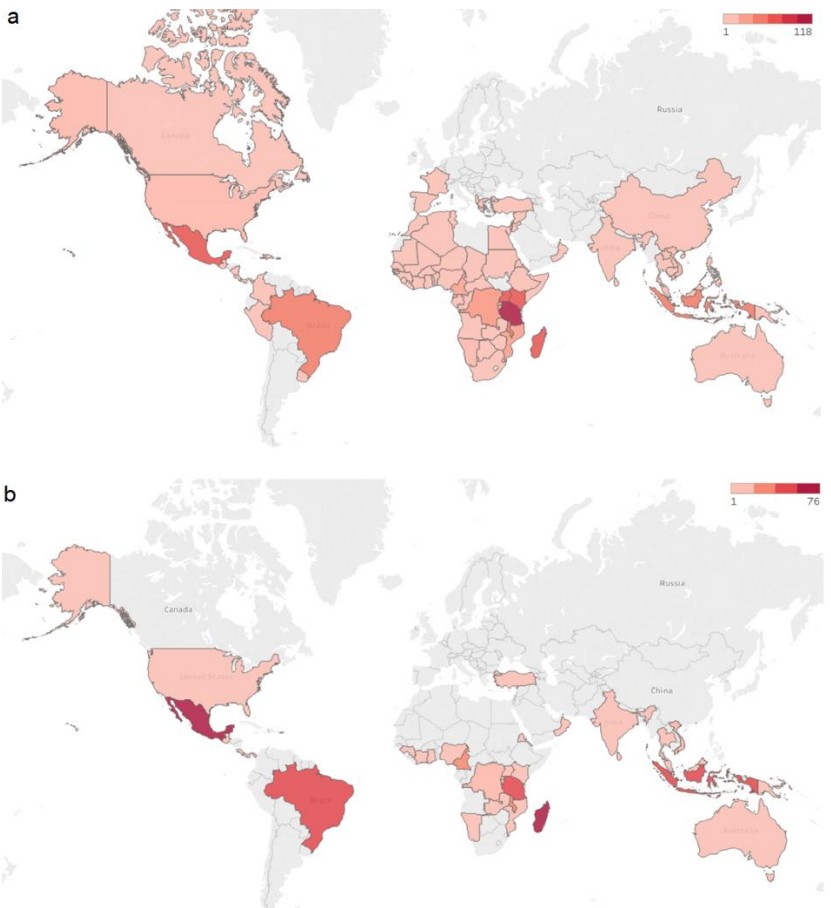

**Figure 2.** Total (**a**) number of species and (**b**) endemics in the CARES priority list. Tableau public 2019.1.

## 3. Discussion

The CARES priority list currently contains nearly six hundred species of freshwater fish from twenty families, including over two dozen extinct-in-the-wild species. Unsurprisingly, the major families in CARES were the most popular ornamental fishes and the ones with the largest affiliations in aquarium hobby organizations such as killifish (Cyprininodontidae), livebearers (Goodeidae, Poeciliidae), as well as cichlids (Cichlidae) [12], with the latter representing nearly half of all species on the CARES list. The over representation of cichlids can be explained by their popularity as freshwater aquarium fish, as well as the fact that they are one of the largest taxonomically diverse and most endangered vertebrate groups due to their endemism in areas facing large anthropogenic pressures. Their high species number is due to their rapid speciation in isolated lakes especially in the East African Rift such as Malawi, Tanganyika, and Victoria which are some the richest and biodiverse lakes in the world [18]. Pseudomugilidae and Valenciidae contained every species listed as threatened by IUCN including an extinct-in-the-wild species for the former and both critically endangered species in the latter. Meanwhile, only around one percent of all the threatened IUCN Characidae, Cobitidae, and Gobiidae species were represented in CARES. Moreover, although Loricariidae represents the largest group of catfishes, none were found in the IUCN red list and over half were not found in fishbase.org. This is because the popularity of armored catfishes in the aquarium hobby has not caught on with the scientific literature due to the fish occurring in areas difficult for studies to be undertaken, with many species designated with an "L-number" instead of a scientific name as most have not been taxonomically described. Nevertheless, all Loricariidae species in CARES are described in the Planet Catfish database, with most found in Brazil, specifically the Xingu River, which has the most sought-after species by hobbyists.

Lake Victoria and Lake Malawi in East Africa are the major locales where many of the CARES species are located, as they are some of the most biodiverse lakes in the world with very popular species in the aquarium trade. Due to the large number of freshwater fish, especially in those two lakes, the Afrotropical region has by far the most threatened freshwater fish species listed in both CARES and the IUCN [19]. Tanzania contained the largest number of CARES species as they are a major exporter for Lake Malawi and Lake Tanganyika cichlids. Three other East African countries, Madagascar, Kenya, and Uganda, contained the third to fifth largest numbers, respectively. Madagascar has nearly half of all their species threatened [20], and the second largest number of endemics in the CARES priority list. Mexico has the second most CARES species, which is no surprise given that it is one of the top five countries with the most threatened species and the one with the most endemics [19]. However, it may also be due to the work of major hobbyist organizations in this country, such as the Goodeid Working Group which supports major Goodeid conservation projects in Mexico such as captive rearing, research funding, habitat restoration, and public outreach [12]. However, CARES did not proportionally represent the top two countries with the most threatened freshwater fish species: the U.S. and India, respectively [19]. The lack of U.S. species presence on this list is not surprising when one considers CARES is predominately a U.S. based organization and the possession of some of these species would be illegal, with the threatened status of most species likely to be known. However, the North American Native Fish Association (NANFA) promotes breeding and the keeping of certain rare North American freshwater fish and could be a useful organization to collaborate with CARES. India harbors the greatest number of endemic species in continental Asia, and the Indomalayan region has the second most threatened freshwater species [19]. Although India is one of the largest fish exporters along with the Philippines [21], recent crackdown on ornamental fish exports has likely increased the difficulty of obtaining many of these species [11]. We recommend CARES should focus more on species from India and other Asian regions, along with Central and South America, which are at exceptional risk due to amount of undescribed species and high extinction risk [22,23]. Nevertheless, some regions and countries will always be better represented than others due to trade restrictions, for example, species from South American countries like Peru and Colombia are easier to obtain than those from Guyana and Surinam. Additionally, we found CARES species were biased towards the tropics and warmer

climates, with few species located in colder and northern countries. We therefore also suggest CARES and similar hobbyist organizations expand to include European and other underrepresented regions.

This study illustrates the discrepancies between aquarium hobbyist organizations and the scientific community. We found CARES listed over eighty species that are currently undescribed by the scientific community, given that they have not been identified in IUCN or fishbase.org, including half of all Loricariidae and a quarter of all Cichlidae. CARES also classified a large percent of red list species differently than the IUCN. Although many classifications were different by just one level, some disparities were much larger, such as in the Anabantidae group where more species were listed as threatened in CARES than in the IUCN. Remarkably, a third of all extinct-in-the-wild CARES species were classified as least concern by the IUCN. The vast disconnect in information of so many species exemplifies the lack of collaboration between hobbyists, the scientific community, zoo/public aquariums, and conservation organizations and the need for stronger partnerships between all these groups to ensure no species is left without proper management. This is highlighted by the fact that many species remain undescribed by the IUCN, with their conservation status subsequently unknown, yet many have been named and are well known by aquarium hobbyists. Around nearly a quarter of all fish species are only found to members of hobbyist organizations, which maintain extensive lists of names, origins, and technical reports of many unclassified and undescribed species [12]. Aquarium hobbyists often possess discrete knowledge based on field observations, while many hobby associations are dedicated to specific fish groups or regions. These detailed descriptions of undescribed species and their physical attributes, ecology, habitats, and where they are found can provide a detailed background for future work by professionals. In the case of CARES, professional scientists involved within the organization can confirm these descriptions and provide a viable starting point for conservation scientists to assess these undescribed species. Collating the diverse streams of knowledge from hobbyists with conservation organizations like the IUCN can help provide more detailed, accurate, and complete species listings.

The main goal of CARES is to maintain populations of endangered or extinct-in-the-wild species through captive rearing to preserve the species for future generations, while also using their knowledge and captive species for conservation and possible reintroductions. Although it is unusual to have non-scientists maintaining endangered species due to the threat of overharvesting and further decimation of these populations, over 90% of freshwater fish in the trade market are captive-bred, which means there is little risk of this occurring [24]. For some species, this market has even saved species from extinction through in-situ and ex-situ conservation [12], producing a source of surplus of individuals required for reintroduction programs [16]. Although CARES involves individuals with scientific backgrounds, there must be closer collaborations with scientists across disciplines as well as partnering with in-situ and ex-situ conservation efforts. Collaborating with ichthyologists and taxonomists can help confirm species identifications; building further relationships with ecologists, conservation biologists, and wildlife management can help in-situ conservation efforts such as habitat conservation, fisheries/population management, reintroductions, and community-based management; and developing partnerships with commercial ornamental aquaculture and the pet industry can help build genetically valuable breeding stock. Moreover, working collaboratively with well-funded and knowledgeable private, governmental, and non-profit conservation organizations, CARES can become a major catalyst for the improved success of many of these conservation programs. For example, partnering with zoos and aquariums can help with ex-situ conservation efforts since fish hobbyists keep many more species than public aquariums while also possessing invaluable information on a species habitat and breeding requirements. A quick search on the AZA website shows there are currently only seven freshwater Actinopterygii fish in conservation action plans, in comparison to the hundreds of species in CARES. Although hybridization is more relevant to the commercial aquarium trade and is looked down upon by hobbyists, we note that CARES conservation may be more beneficial to highly endemic fish species as it is much easier to avoid accidental hybridization. Nevertheless, zoological organizations such as AZA and EAZA also have species survival plans which can help to

maintain genetic diversity in captive populations. Ultimately, we stress the importance of scientists and conservation organizations to recognize organizations such as CARES as a valuable resource to help fill in the knowledge gaps and provide a mutualistic exchange of knowledge.

Aquarium keeping is rated as the second most popular hobby with millions owning ornamental fishes such as guppies, bettas, and goldfish [17]. Due to their interest in fish conservation and a love for the species they own, many aquarium hobbyists can often be more knowledgeable than the scientific community while often caring and breeding species that are critically endangered or already extinct-in-the-wild. Aquarium keeping in general also helps form a place attachment and can bring awareness to specific areas and ecosystems as individuals try to recreate a particular biotope for their fish tank. It helps people care about threatened and endangered places that they have never been to and countries they have never visited, with some hobbyist inspired by their tanks to partake in ecotourism to go visit these places. Aquarium organizations help bring passionate hobbyists together to exchange information to maintain and preserve specific groups of fishes and their habitats. CARES combines many of these organizations to specifically breed and keep species that may soon be and in some cases already are gone in the wild. The importance of programs such as CARES is that they focus on many species with little to no commercial value in the ornamental fish trade and emphasize those which are overlooked and not considered charismatic enough for zoos and aquariums. However, to fulfil their goals, there must be closer partnerships with the scientific community, large well-funded organizations such as the AZA/EAZA, and conservation organizations such as the IUCN. By bridging these gaps, fish hobbyists through CARES can help play a major role to help preserve fishes for future generations.

## 4. Materials and Methods

The CARES priority list was downloaded from https://caresforfish.org on 21 May 2019, which included nomenclature and risk classification from IUCN and CARES authorities. Fish species data was downloaded from the IUCN red list to compare classification risk and determine what groups are well represented [19]. We used the "rfishbase" package [25] in R version 3.6.0 [26] to find species that could not be found in the IUCN red list by searching for synonyms and other possible names, as well as find the countries and regions the species are found. This package accesses the fishbase database (http://www.fishbase.org) which describes ecology and biology of the over 30,000 known fish species. We also accessed online databases such as Cat-eLog (https://www.planetcatfish.com/catelog/), seriouslyfish (https://www.seriouslyfish.com/), and the Goodeid Working Group database (http://www.goodeidworkinggroup.com/) to find information of CARES species still not found in the previous databases.

**Author Contributions:** Conceptualization, J.V. and K.M.; methodology, formal analysis, writing, visualization, J.V; review and editing, K.M.

**Funding:** This research received no external funding.

**Acknowledgments:** The authors would like to dedicate this publication to Dr. Raydora Drummer Francis for her contributions and support to minority students in the STEM fields. We thank her for helping us and so many others successfully navigate through our academic and personal lives while inspiring us to become the people we are today. We also wish to thank John Gould for his wonderful work proof-reading and editing.

**Conflicts of Interest:** The authors declare no conflict of interest.

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
