# Peer review of "Assessing the Species in the CARES Preservation Program and the Role of Aquarium Hobbyists in Freshwater Fish Conservation"

_fishes, doi:10.3390/fishes4040049_

Round 1

Reviewer 1 Report

This manuscript summarizes the CARES initiative of hobbyists to keep endangered or even extinct-in-the-wild fish species in captivity. It further outlines that species numbers referenced by CARES are much larger than those covered by the UICN red lists. The manuscript makes important statements on the vital and necessary but often overlooked role of non-scientific fish keeping for the conservation of fish biodiversity. The text is very well written and it was a pleasure to read especially for me as a person that scientifically works with fishes and also enjoys having them in an aquarium at home. Nevertheless, I think the authors present an overly positive picture of CARES and similar programs that are not under strict scientific supervision. Thus, I would like to ask the authors to include some discussion on the following points:

What about misidentification of species? How sure are we that hobbyists know what species they take care of? For many rare (and often endangered) species this is not a trivial point and while scientists may control each other through peer-review of published articles, hobbyists typically lack these kinds of ‘quality insurance’.

As a consequence of the above mentioned point: How do we deal with unintended hybridization, a topic that is relevant for many freshwater species especially poeciliids and cichlids that often interbreed? Is there any guarantee mechanism that prevents hybridization or at least identifies when it has happened?

In opposition to the authors’ points at lines 235, I assume over-fishing can be a problem especially for rare, endemic species that are not easily reproduced in captivity. Often, rare wild-caught species (i.e., L-catfishes) are sold at high prizes which can facilitate their exploitation.

These points are by no means a cause that should prevent the publication of this article after a revision.

Author Response

This manuscript summarizes the CARES initiative of hobbyists to keep endangered or even extinct-in-the-wild fish species in captivity. It further outlines that species numbers referenced by CARES are much larger than those covered by the UICN red lists. The manuscript makes important statements on the vital and necessary but often overlooked role of non-scientific fish keeping for the conservation of fish biodiversity. The text is very well written and it was a pleasure to read especially for me as a person that scientifically works with fishes and also enjoys having them in an aquarium at home. Nevertheless, I think the authors present an overly positive picture of CARES and similar programs that are not under strict scientific supervision. Thus, I would like to ask the authors to include some discussion on the following points:

We wish to thank the reviewer for their time and consideration.

What about misidentification of species? How sure are we that hobbyists know what species they take care of? For many rare (and often endangered) species this is not a trivial point and while scientists may control each other through peer-review of published articles, hobbyists typically lack these kinds of ‘quality insurance’.

We agree with the reviewer about the concern for species mis-identification. The most accurate way for the identification of a species is through a very detailed description of species characteristics, such as those provided by hobbyists within their respective organizations. For example, although mostly undescribed by science, the L numbers in catfishes are all backed up by very detailed descriptions of physical attributes, ecology, habitats, and where they are found, all which can provide a background for future work by professionals. In the case of CARES, the organization includes professional scientists which can confirm these descriptions. However, one of the major aims of this study was to emphasize the importance collaboration with other scientists and organizations, as stated in the discussion. We revised the introduction and discussion to emphasize these points. 

As a consequence of the above mentioned point: How do we deal with unintended hybridization, a topic that is relevant for many freshwater species especially poeciliids and cichlids that often interbreed? Is there any guarantee mechanism that prevents hybridization or at least identifies when it has happened?

Although there is always a possibility of hybridization it is more relevant to the commercial aquarium trade. Hybridization is looked down upon within hobbyists and their respective organizations, especially when it involves conservation such as CARES. Species from groups such as cichlids which may have a large range are typically bred with their location in mind to avoid hybridization. Nevertheless, this remains a possibility and is why we suggest stronger collaboration and partnerships with conservation organizations and scientists. We added to the discussion that CARES is more likely beneficial to highly endemic fish species because it is much easier to not accidentally hybridize. 

In opposition to the authors’ points at lines 235, I assume over-fishing can be a problem especially for rare, endemic species that are not easily reproduced in captivity. Often, rare wild-caught species (i.e., L-catfishes) are sold at high prizes which can facilitate their exploitation.

By commercial value, we don't necessarily mean an individual is worth a lot of money. Although rare endemic species may be expensive, it may not have much commercial value if there are not too many people interested in the species. Fish of commercial value may not not be relatively expensive but can be scaled to a large number by demand. In some cases, rare fish like the Asian Arowana can have large commercial value and can result in their exploitation because of high demand. As far as we know, although L-number catfish have a high price it has not declined due to trade but mostly due to habitat loss and building of dams. However, the aim of organizations such as CARES is specifically for fish species that are threatened but are with little to no commercial value in the ornamental fish trade, and emphasize those which are overlooked and not considered charismatic enough for zoos and aquariums. For example, species such as Skiffia francesae is has no value commercially as it is not an attractive fish and was never collected in high numbers. However, it is extinct in wild due to the invasive and commercially valuable platyfish. This species exists only because it was specifically saved by hobbyists. We expanded the discussion to address these points.

These points are by no means a cause that should prevent the publication of this article after a revision.

Reviewer 2 Report

The paper by Valdez and Mandrekar discuss the CARES program, which is made up of hobby aquarist organisations with an aim of preserving the threatened freshwater fish species available in the hobby. In particular, the paper includes an analysis of the overlap between the CARES priority list and the IUCN Red List.

With respect to the analysis, there is not much to argue with. Is is a simple basic comparison, and the authors basically describes the overlap and the differences in the two lists. However, I wonder how much information this actually adds to ichthyology in general. Both lists are available online, and the comparison will be outdated as soon as any of the lists change. Nevertheless, it may be useful to spread the message about the efforts that are made in preserving fish species from the aquarist hobbyists.

There are some statements that may be inappropriate in the paper. For instance, the claim that freshwater fishes are the most threatened vertebrate group in the world (L 38). ”Freshwater fish” is not a taxonomic group of vertebrates. Since the word ”group” is used without definition, it is easy to find other ”groups” of vertebrates that are more threathened – ”apes in Borneo” may be a group that on average is more threathened than ”freshwater fish”. We should also recognise that there are thousands of freshwater fish species that are not threathened at all…

The extinction rate of 877 times the background rate seems to originate from an article that only concern North America. Overall, I believe that the authors need to tone down the claims on how vulnerable ”freshwater fishes” are. The main reason is that there is so much variation in this ”group” (in terms of species diversity and threat levels) that it makes little sense writing about them as a single group that should be considered as a unit of concern.

L74-75: No reference is given for the statment about ornamental fish trade.

L93 and onward: Scientific names should be italicised.

L105: What is the genus for the Potosi pupfish?

Figure 2: The authors might want to discuss that the reason for why most threathened fish in the CARES list comes from a few particular countries, is likely a result of these countries being the main source of aquarium fishes. There are e.g. very few specis from Europe that are kept as aquarium fishes at all.

L196: None of the species listed here (discus, oscar and angelfish) are actually in the list – it ma be more relevant to mention the Great Rift Lake cichlids, which are also common aquarium species, and also in the actual CARES list. Over-representation of cichlids in the list is likely due to these particular fish being both extremely taxonomically diverse and threathened because they are found in a small geographic area which faces a lot of anthropogenic pressure.

L231: A reference to a biblical fairytale (Noah’s Ark) seems inappropriate in a scientific paper.

It should be acknowledged that the IUCN Red List is not yet complete. More species are added continuously. Many fish groups have not yet been assessed, which may explain why some species are missing from that list.

The authors should, in addition to the databases they already used, also cross-reference names to Eschmeyer’s Catalog of Fishes (http://researcharchive.calacademy.org/research/ichthyology/catalog/fishcatmain.asp). This is one of the main sources for taxonomical information in fishes, which provides a lot of information about name validity, synonyms etc.

Author Response

The paper by Valdez and Mandrekar discuss the CARES program, which is made up of hobby aquarist organisations with an aim of preserving the threatened freshwater fish species available in the hobby. In particular, the paper includes an analysis of the overlap between the CARES priority list and the IUCN Red List.

We wish to thank the reviewer for their time and consideration.

With respect to the analysis, there is not much to argue with. Is is a simple basic comparison, and the authors basically describes the overlap and the differences in the two lists. However, I wonder how much information this actually adds to ichthyology in general. Both lists are available online, and the comparison will be outdated as soon as any of the lists change. Nevertheless, it may be useful to spread the message about the efforts that are made in preserving fish species from the aquarist hobbyists.

This study adds to ichthyology and conservation biology in general by illustrating and bringing to light how lay people have taken grass roots initiatives and have bred fish of noncommercial value to save them from extinction. The purpose was to show that such groups can add value to conservation and fill in gaps left by conservation organizations, zoos, and scientists. We conducted a general analysis on CARES and illustrated specific gaps which can be filled with collaboration with other organizations and researchers and provided broad suggestions for future directions to move further. 

There are some statements that may be inappropriate in the paper. For instance, the claim that freshwater fishes are the most threatened vertebrate group in the world (L 38). ”Freshwater fish” is not a taxonomic group of vertebrates. Since the word ”group” is used without definition, it is easy to find other ”groups” of vertebrates that are more threathened – ”apes in Borneo” may be a group that on average is more threathened than ”freshwater fish”. We should also recognise that there are thousands of freshwater fish species that are not threathened at all…

Although not a taxonomic group, they are considered by the IUCN one of the most threatened group by the total number of species.  However, this wasn't made clear what was meant by most threatened. We have changed it to "the most threatened vertebrate group by number of species"

The extinction rate of 877 times the background rate seems to originate from an article that only concern North America. Overall, I believe that the authors need to tone down the claims on how vulnerable ”freshwater fishes” are. The main reason is that there is so much variation in this ”group” (in terms of species diversity and threat levels) that it makes little sense writing about them as a single group that should be considered as a unit of concern.

We agree with the reviewer and removed that statement. We also agree that there is much variation within freshwater species and they contain diverse taxonomic groups. However, we refer to them as a group based on the IUCN which recognized a need for their conservation and established the Freshwater Fish Specialist Group to examine this group of fish due to their vulnerability from anthropogenic threats to their habitats. 

L74-75: No reference is given for the statment about ornamental fish trade.

References have been added.

L93 and onward: Scientific names should be italicised.

Names have been italicized. 

L105: What is the genus for the Potosi pupfish?

Genus has been added.

Figure 2: The authors might want to discuss that the reason for why most threathened fish in the CARES list comes from a few particular countries, is likely a result of these countries being the main source of aquarium fishes. There are e.g. very few specis from Europe that are kept as aquarium fishes at all.

We discussed the reasons for this in the second paragraph of the discussion. Hobbyists seem to be more biased towards warmer and tropical fish species, which is why the only European species were in Iberia and the Mediterranean. We have added this to the discussion.

L196: None of the species listed here (discus, oscar and angelfish) are actually in the list – it ma be more relevant to mention the Great Rift Lake cichlids, which are also common aquarium species, and also in the actual CARES list. Over-representation of cichlids in the list is likely due to these particular fish being both extremely taxonomically diverse and threathened because they are found in a small geographic area which faces a lot of anthropogenic pressure.

We agree with the reviewer and have revised as suggested.

L231: A reference to a biblical fairytale (Noah’s Ark) seems inappropriate in a scientific paper.

We agree with the reviewer and have removed the statement.

It should be acknowledged that the IUCN Red List is not yet complete. More species are added continuously. Many fish groups have not yet been assessed, which may explain why some species are missing from that list.

The authors agree with the reviewer that the IUCN red list is not complete and will continuously be updated. Many fish groups and species have yet to be assessed and one of the main goals of this paper is to demonstrate that organizations such as CARES contain many unlisted species which can provide a viable starting point when assessing new species.We explained why some species are missing from the list in the abstract and discussion "...the popularity of armored catfishes in the aquarium hobby has not caught on with the scientific literature due to the fish occurring in areas difficult for studies to be undertaken, with many species designated with an “L-number” instead of a scientific name as most have not been taxonomically described. "

The authors should, in addition to the databases they already used, also cross-reference names to Eschmeyer’s Catalog of Fishes (http://researcharchive.calacademy.org/research/ichthyology/catalog/fishcatmain.asp). This is one of the main sources for taxonomical information in fishes, which provides a lot of information about name validity, synonyms etc.

We thank the reviewer for the suggestion of the database. We cross-referenced for the undescribed species in our dataset, but could not identify any of the missing species. This is likely because, as previously stated in the discussion, the missing species are undescribed by the scientific community but were designated a name by the fish trade and hobbyist organizations based on physical characteristics and location. 

Reviewer 3 Report

The Ms is addresses a very interesting and relevant topic and relevant for the conservation of biodiversity of freshwater fish. It reads very well, is adequately organized (just a minor comment, please see below). The literature is relevant, adequate for the topic and well handled.

I have only two very minor comments:

Species names in Latin must be presented in italics and the section Materials and Methods must be included after the Introduction and before the Results section. This will contribute to a more adequate organization of the Ms.

Author Response

The Ms is addresses a very interesting and relevant topic and relevant for the conservation of biodiversity of freshwater fish. It reads very well, is adequately organized (just a minor comment, please see below). The literature is relevant, adequate for the topic and well handled.

The authors wish to thank the reviewer for their time and consideration.

I have only two very minor comments:

Species names in Latin must be presented in italics and the section Materials and Methods must be included after the Introduction and before the Results section. This will contribute to a more adequate organization of the Ms.

Latin names have been italicized. We agree with the reviewer, but we included the Results before Materials and Methods as required by the format of the journal.